# Effects of Nutrient Intake during Pregnancy and Lactation on the Endocrine Pancreas of the Offspring

**DOI:** 10.3390/nu11112708

**Published:** 2019-11-08

**Authors:** Valentine Suzanne Moullé, Patricia Parnet

**Affiliations:** Nantes Université, INRA, UMR 1280, PhAN, IMAD, F-44000 Nantes, France; patricia.parnet@univ-nantes.fr

**Keywords:** dietary intake, endocrine pancreas, gestation, lactation, offspring

## Abstract

The pancreas has an essential role in the regulation of glucose homeostasis by secreting insulin, the only hormone with a blood glucose lowering effect in mammals. Several circulating molecules are able to positively or negatively influence insulin secretion. Among them, nutrients such as fatty acids or amino acids can directly act on specific receptors present on pancreatic beta cells. Dietary intake, especially excessive nutrient intake, is known to modify energy balance in adults, resulting in pancreatic dysfunction. However, gestation and lactation are critical periods for fetal development and pup growth and specific dietary nutrients are required for optimal growth. Feeding alterations during these periods will impact offspring development and increase the risk of developing metabolic disorders in adulthood, leading to metabolic programming. This review will focus on the influence of nutrient intake during gestation and lactation periods on pancreas development and function in offspring, highlighting the molecular mechanism of imprinting on this organ.

## 1. Introduction

The importance of well balanced, high quality nutrition during gestation and lactation periods is essential to the proper development of the fetus and the baby. In fact, at birth, the majority of mammals are not fully developed, and the lactation period is a time of accelerated growth for most organs. For example, the rate of brain growth accelerates during the last trimester in humans, culminating at birth, but occurs relatively late in the rat, with a peak around postnatal days (PND) 7 to 8 [1]. In humans, new pancreatic beta cells are formed between birth and the fully developed young adult [2] and it has been shown in several animal models that neogenesis accounts for an important part of the new pancreatic beta cells in rodents [3,4,5].

In recent decades, perinatal nutrition has become a key player in susceptibility to metabolic disorders in adults. Epidemiological and experimental studies have shown that low birth weight (LBW) resulting from slow intrauterine growth increases the risk of developing metabolic disorders in adulthood, giving rise to the concept of the developmental origin of health and disease, or “metabolic programming” [6]. It is now well recognized that there is a relationship between birth weight, rapid catch-up growth and adult fat mass, with low or high birth weight associated with a higher prevalence of developing obesity at adulthood [7]. In addition, a lower body weight at birth is associated with multiple pathologies that define the metabolic syndrome: hypertension, dyslipidemia, hyperinsulinism, impaired glucose tolerance [8].

In this review, we focus on the effects of maternal dietary intake during pregnancy and lactation on offspring glucose metabolism and pancreas structure. Nowadays, eating habits have changed with the availability of ultra-processed foods and the emergence of new specific diets, such as veganism or gluten-free diet. Food quality also varies widely, with the consumption of more fat, sugar, salt or preservatives. As the pancreas continues to develop after birth, it is important to understand the impact of these new dietary challenges on its development and the consequences on glucose metabolism in adulthood.

## 2. Pancreas 

The endocrine pancreas is a key player in the regulation of blood glucose levels by secreting insulin, glucagon, somatostatin and pancreatic polypeptide into the blood. Its exocrine part, by the secretion of digestive enzymes in the duodenum, plays an important role in digestion. We will focus here on the anatomy, development and function of the endocrine part.

### 2.1. Pancreas Anatomy

The pancreas is a long, slender organ—most of which is at the bottom of the stomach in humans. In rodents, it is a diffuse organ located between the spleen, stomach, intestine and colon. The pancreas is connected to the small intestine through the bile duct and receives blood through the pancreatic artery.

The exocrine tissue is responsible for digestion, particularly lipid degradation through bile secretion (Figure 1). The exocrine part is organized by acini which regroup acinar cells that secrete zymogens for digestive hormones in pancreatic ducts. The gall bladder and pancreatic duct release digestive enzymes and bile in the duodenum. Pancreatic duct cells also participate in the neutralization of stomach acid in the duodenum by secreting bicarbonate [9]. 

Embedded within the exocrine tissue, endocrine cells are organized into islets containing beta, alpha, delta, pancreatic polypeptide and epsilon cells respectively secreting insulin, glucagon, somatostatin, pancreatic polypeptide, and ghrelin. The endocrine pancreas represents only 1%–2% of pancreatic tissue despite its essential role in the regulation of the metabolism throughout life. Islet architecture differs between humans and rodents [10]. In humans, endocrine cells are distributed in islets with no particular order. However, in rodents, beta cells are located in the core of islets surrounded by other endocrine cells. The percentage of beta cells is also larger in rodent islets than in human islets.

The exocrine and endocrine parts of the pancreas are innervated by the sympathetic nervous system (SNS) and the parasympathetic nervous system (PNS)—the two branches of the autonomic nervous system. The main neurotransmitters released by postganglionic sympathetic neurons are norepinephrine, galanin and neuropeptide Y. SNS acts to maintain glycemic levels during stressful situations by decreasing insulin and increasing glucagon secretion, but its role on exocrine pancreases remains controversial [11]. The PNS acts in two different ways, with specific neurotransmitters: the excitatory cholinergic pathway involving the release of acetylcholine and the non-adrenergic non-cholinergic inhibitory pathway involving nitric oxide, vasointestinal peptide (VIP), gastrin-releasing peptide (GRP) and pituitary adenylate cyclase-activating polypeptide. Contrary to the SNS, the PNS stimulates insulin release and plays a role during the cephalic phase of insulin release which is defined by the release of gut hormones and digestive enzymes before the systemic hormonal response induced by the ingested nutrients. It also plays an important role on pancreatic exocrine secretions, whose peptide secretions such as cholecystokinin or pancreatic polypeptide are vagally mediated. Finally, the innervation of the pancreas varies according to the species [10]. In the mouse, the sympathetic axons are located on the periphery of the islets, near the alpha cells, whereas in man, these types of axons are rare and generally localized around the blood vessels, inside the islets. With regard to the PNS, the islets of mice seem to be strongly innervated inside the islet, while the human islets are barely innervated.

### 2.2. Development and Plasticity of the Pancreas

The anatomy of the fully developed pancreas depends on a complex sequence of prenatal and postnatal processes [3,4,5,9] under epigenetic influences (see for review [12]).

In the mouse, the embryonic day (E) 8.75 marks a turning point for the development of the pancreas [9] and this takes place immediately after the closure of the anterior endoderm. In humans, the first signs of pancreas formation appear on gestation day 26 (G26d) [5]. The expression of pancreatic and duodenal homeobox 1 (Pdx-1) eventually increases around G29–30d. The coexpression of Pdx1, homeobox HB9 (H1xb9), pancreas-associated transcription factor 1a (Ptf1a), homeobox 1 NK6 (Nkx6-1) and homeobox protein Nkx-2.2 (Nkx-2.2) also defines common pancreatic progenitor cells in the epithelium together with Nkx6-2 and SRY-Box 9 (Sox9) in mouse [9]. Almost the same markers are found in humans, suggesting a significant similarity in terms of molecular markers of early pancreatic differentiation [5]. Glucagon and Insulin are detected from E9.5 in mice, with glucagon being the most abundant [9]. In humans, these two hormones are detectable around the 7.5 week of gestation, but insulin is synthetized first and predominantly [5].

At birth, human and rodent pancreases are not mature and will undergo several modifications to reach maturity. Rodent growth is very significant during the first postnatal month as well as for the pancreas which develops rapidly between 2 and 4 weeks after birth [4]. Both exocrine and endocrine parts of the pancreas increase. The extension of pre-existing lobes and the formation of new lobes by proliferation of the ducts are accompanied by their differentiation in islets and acini. Before weaning, neogenesis from undifferentiated cells participates to the beta-cell mass increase. During weaning, the balance between beta-cell apoptosis and proliferation determines the transformation of the fetal pancreas into its post-weaning state. After weaning, the plasticity and turnover of islets and beta cells are then reduced, strengthening the importance of an optimal prenatal and postnatal pancreatic development [13,14]. In humans, an increase in islet size instead of islet number is responsible for beta-cell mass expansion from birth to adulthood [2]. The postnatal expansion of beta-cell numbers is largely due to replication of existing beta cells, with the greatest rate during infancy. The capacity of beta cell replication and proliferation dramatically declines with age, being close to zero from approximately 5 years old. Beta-cell apoptosis has been detected in human pancreas but at very low levels, suggesting that this phenomenon does not apply to pancreas remodeling after birth [2]. The study of the postnatal development of human islets is limited to the static observation of postmortem pancreas samples and is dictated by ethical issues regarding fetal samples. 

Newly formed beta cells have a limited capacity to secrete insulin in young individuals, both rodents and humans. Beta cells have a greater response to nutrients other than glucose [15]. This is explained by the absence or the weak expression of key genes involved in glucose metabolism, ATP amplifying pathways and insulin secretion in young beta cells. For example, insulin or glucose transporter-2 (GLUT-2) expression reach adult levels around PND28 [16]. Since the number of beta cells increases in the first postnatal month, the secretory immaturity of newly formed beta cells could hide the robust secretion from a more mature population of beta cells [4]. In humans, very few studies were interested in insulin secretion in early age. As in rodents, the insulin secretory function is completely immature in islets from a 5-day-old donor [17]. The pharmacological increase in cyclic adenosine monophosphate levels by using forskolin tends to improve the profile of glucose-stimulated insulin secretion (GSIS) compared to older donors between 11 and 36 months old [18]. 

The number of beta cells and their ability to secrete insulin depend on physiological conditions [19]. Among them, pregnancy has a significant impact on maternal metabolism and insulin secretion in order to provide optimal nutritional supply to the fetus. The third trimester in humans and the last days of gestation in rodents are characterized by marked insulin resistance, leading to a greater insulin secretion in response to glucose and an increase in beta-cell mass with beta-cell hypertrophy and hyperplasia [20]. After delivery, beta-cell apoptosis occurs, resulting in a reduction of beta-cell volume and lower cell replication [21]. The pathophysiological state of obesity is also characterized by the adaptation of beta cells. Obesity is a risk factor for type 2 diabetes (T2D) and despite the fact that obese people are often glucose tolerant, insulin resistance is often diagnosed in this population [22]. High levels of circulating insulin contribute to increased individual beta-cell activity, but also promote their proliferation, resulting in an increase in beta-cell mass [23,24]. More recent findings have also shown, both in vivo and in vitro, the specific role of excess glucose and/or circulating lipids on beta-cell proliferation, independently of insulin levels [14,25].

### 2.3. Regulation of Insulin Secretion

The endocrine pancreas is essential in maintaining normal blood glucose levels with the help of insulin, the only hypoglycemic hormone. Its secretion can be stimulated by an increase in nutrients, primarily glucose, or by activation of the SNS. The release of insulin in response to glucose is biphasic as shown in vivo and in vitro: a rapid and transient initial peak of 3–10 minutes followed by a second slowly developing release phase [26]. This specific release scheme is explained by the rapid discharge of an insulin pool present in the granules anchored in the plasma membrane, followed by the slower release of a pool of granule reservoirs located further.

#### 2.3.1. Glucose-Stimulated Insulin Secretion

In cells, glucose is transported by the low-affinity, high-capacity glucose transporter-2 (GLUT-2), which allows rapid uptake of glucose regardless of extracellular sugar concentration. GLUT2, also called solute carrier family 2, member 2 (SLC2A2) is a facilitative glucose transporter located in the plasma membrane of the liver, pancreatic, intestinal, kidney cells as well as in the portal and the hypothalamus region. Glucose is rapidly phosphorylated once in the cell by the rate-limiting enzyme glucokinase. This low affinity enzyme allows significant variation in activity within the range of physiological glucose concentrations [27]. Intracellular glucose metabolism generates multiple ATP molecules, increasing the ATP/ADP ratio and promoting the closure of ATP-sensitive K+ channels. The closure of these channels causes the depolarization of the plasma membrane and the opening of voltage-dependent Ca^2+^ channels. The rise in intracellular Ca^2+^ triggers several pathways that reinforce insulin exocytosis and contribute to the regulation of beta-cell proliferation [27,28]. 

Part of the secretion of insulin is controlled by the central nervous system, which plays a major role in the cephalic phase of insulin release (i.e., first phase). This corresponds to the pre-absorptive secretion of insulin in response to neural signals triggered by meal olfactory clues rather than changes in plasma glucose concentrations after meal intake [29,30]. Between the two components of the autonomic nervous system, the PNS is the one that carries out a significant role in regulating insulin secretion. This mainly involves the release of acetylcholine and activation of muscarinic receptor 3 located on the beta cells [11,30]. However, non-cholinergic mechanisms mediated by neuropeptides such as VIP and GRP have also been reported in human and pigs. However, it is still unclear to what extent non-cholinergic mechanisms are involved in hormonal secretion during the cephalic phase. On the other hand, the sympathetic innervation is unlikely to affect insulin secretion because it mainly inhibits insulin secretion during hypoglycemia state [11].

#### 2.3.2. Other Contributors to Insulin Secretion

Several other signals can trigger intracellular pathways and enhance GSIS [28]. At the surface of beta cells, G-coupled protein receptors mediate the effects of numerous regulators of insulin secretion such as neurotransmitters (e.g., acetylcholine, noradrenaline), fatty acids (e.g., oleate, palmitate) and hormones (e.g., somatostatin, glucagon-like peptide 1 (GLP-1)). They generate metabolic coupling factors involved in intracellular calcium increase and insulin exocytosis. Amino acids such as glutamine and leucine can enhance GSIS. Other amino acids indirectly participate in insulin release via their capacity to increase blood levels of glucagon, GLP-1 and gastric inhibitory peptide (GIP) [31].

## 3. Influence of Dietary Intakes during Gestation and or Lactation

Several studies have underlined the impact of nutrition during gestation and lactation periods on offspring fate. A common consequence observed in the context of nutrient excess or restriction is the imbalance of specific amino acids or fatty acids which could create competition for enzymes or transporters, reducing placental growth or nutrient supply to the fetus. Here, we will review the influence of nutrient excess or nutrient restriction, during these critical periods, on glucose homeostasis regulation in offspring.

### 3.1. Excess of Nutrients

In Western countries, overconsumption of high-energy food is pretty common, especially because of plethoric food availability. Associated with increased sedentary behavior, food overconsumption contributes to the deregulation of energy balance in favor of weight gain [32]. The World Health Organization (WHO) estimates that in 2016, over 1.9 billion adults, 18 years and older, were overweight—of which, over 650 million were obese. Regarding children, 41 million under the age of 5 were overweight or obese and over 340 million children and adolescents aged 5–19 were overweight or obese. This greatly increases the risk of developing metabolic syndrome, T2D and other chronic diseases. Because of worldwide development of obesity, the number of obese women during childbearing age increases, leading to a higher proportion of gestational diabetes (GD). Half of the GD prevalence can be explained by overweight and obesity. Today, GD impacts approximately 20% of pregnancies in the world and this is close to 40% for women in Europe [33]. Evidence also links GD and/or obesity to increased risk of maternal and neonatal outcomes such as T2D in mothers and offspring obesity [34,35]. Exposure to an excess of maternal nutrients during the in utero period could also affect a number of metabolic pathways in developing organs, such as the liver, skeletal muscle, adipose tissue, brain, and pancreas, and favor epigenetic modifications, leading to obesity [36,37].

#### 3.1.1. Carbohydrate Excess

Carbohydrate ingestion mainly provides glucose, which is the key source of energy in the body, essential for the brain and red blood cells.

When the pregnant dam consumes a larger quantity of sugar during pregnancy, metabolic changes in the pups may persist during post-natal life. This affects pancreas development and glucose tolerance in male offspring [38]. In addition, when offspring have been exposed to high sucrose during gestation, exposure to high sucrose later in adulthood worsens glucose tolerance. This could be explained by a default of hepatic insulin signaling [38]. Sugar consumption by rat dams during gestation and lactation induces an increase in body weight, blood glucose and plasma insulin levels in mothers and pups in a dose-dependent manner [39]. Sugar consumption also increases histopathological damages in pup pancreas, and this is associated with the decrease in insulin- and glucagon-positive cells and a lower expression of insulin receptors. This study experimentally demonstrates that a high daily intake of sugar in healthy pregnancy causes adverse effects on the mother and offspring [39]. These effects could involve epigenetic changes, since long-term exposure of pancreatic beta cells to high glucose concentrations increases DNA methylation of the *Ins1* promoter in the INS-1 insulinoma cell line and in pancreatic islets isolated from male Zucker diabetic fatty rats [40].

#### 3.1.2. Lipid Excess

Maternal high-fat (HF) feeding emerged as a risk factor for metabolic disorders involving abnormal glucose homeostasis and reduced whole-body insulin sensitivity in offspring [41,42,43]. Studying the impact of lipids on endocrine pancreatic function is a challenging task due to the high chemical heterogeneity of lipid molecules. In fact, the carbon chain length and the level of unsaturation of the lipid structure will confer different proprieties to these molecules. A few examples are given below. High saturated fat intake contributes to insulin resistance, β-cell failure, and TD2, while the consumption of polyunsaturated fatty acids (PUFAs) has anti-inflammatory effects. An enriched saturated fatty acid diet seems to inhibit islet development, whereas a diet enriched in unsaturated fatty acids has the opposite effect, promoting the development of pancreatic islets [41]. The quality of consumed fat has evolved in the last fifty years. Fat origin in food processing has changed, even in the way animals have been fed, leading to a rise in saturated fat consumption and also in *n*-6 PUFA, since saturated fats have been replaced by vegetable oils. This is at the expense of *n*-3 PUFA [44], leading to *n*-6/*n*-3 ratios well above nutritional recommendations. Because *n*-3 and *n*-6 PUFA synthesis pathways utilize common enzymes, an unbalanced consumption of both *n*-3 and *n*-6 precursors can disrupt the synthesis of essential fatty acids such as docosahexaenoic acid (DHA). However, this fatty acid is crucial for the good development of the fetus during pregnancy and after birth [45]. This again highlights the importance of applying a nutrient-balanced diet during pregnancy and the nursing period on the later insulin secretory capacity in the offspring.

From birth, offspring body weight is affected by maternal HF feeding. Indeed several studies have shown an increase in body weight in rodents [42,43,46,47]. However other studies have not observed any changes [48,49,50,51] and some even describe a decrease in pup body weight [41]. The higher body weight at birth does not last during growth [42,46] and the lower body weight at birth persists in adulthood at least in males when offspring are exposed to a junk-food diet during the suckling period and when they are given free access to a range of foods of varying macronutrient composition [51].

The structure of the pancreas is strongly affected by maternal HF feeding. Islet diameter, islet mass, and beta-cell mass are increased by maternal HF feeding in 3-month-old mice offspring [47]. At birth, beta-cell mass is decreased in offspring from obese mothers but returns to a normal level in adulthood because of a higher level of proliferation during post-natal growth [42]. Zambrano et al. showed that the percentage of beta cells is decreased in both male and female offspring from obese mothers without changes in islet morphology at adolescence, which could serve as a compensatory mechanism to improve insulin secretion [48]. On the opposite side, the percentage of alpha cells is increased at this point for both sexes and remains higher only for males in adulthood. Another study showed that islet mass, including beta- and alpha-cell masses, is increased in offspring in the later F1 and F2 generations, suggesting epigenetic modifications in genes involved in pancreatic cell differentiation [12,43]. In vitro, the presence of histone deacetylase inhibitors on embryonic pancreas promote the differentiation of cell precursors in endocrine cells, resulting in increases in both beta- and delta-cell populations [52]. On the contrary, the overexpression of histone deacetylase enzymes decreases these two populations [53].

Modifications of pancreas structure may have consequences on insulin secretion. Ex vivo, islets from female offspring have a stronger response to glucose (5 mM) than islets from male offspring whatever the age (PND36 and PND110) [48]. However, both male and female offspring islets lose their capacity to respond to glucose stimulation when they originate from obese mothers. Dynamic assessment of insulin secretory response to glucose infusion (GSIS) does not show important perturbations of basal insulin secretion in islets originated from female offspring born to mothers fed with HF during gestation and lactation but GSIS is significantly reduced in both first and second phase of insulin release when stimulated with 20 mM glucose [50]. In another model of maternal HF feeding, insulin secretion is altered only in response to palmitate and GLP-1 in islets from male offspring born from dams fed with HF diet during gestation [49]. In vivo, it has been shown that offspring from mothers fed with short-chain saturated fatty acids during gestation and lactation periods present higher plasma insulin levels in response to oral glucose load without differences in terms of glycaemia, suggesting a faint insulin resistance [41]. Furthermore, 8 weeks of maternal HF feeding before mating and gestation expand the deleterious effect of excessive lipids during gestation on glucose tolerance [42,43].

Overall, it seems that there is a strong sexual dimorphism between male and female offspring [48]. Unfortunately, the majority of studies do not evaluate the impact of maternal HF feeding during gestation and lactation on both sexes. In view of the actual limited literature, it appears primordial to investigate more deeply these differences to understand the specific adaptation of each sexes and the potential relation with sex hormones to maternal diet during gestation and lactation. Furthermore, the measure of insulin secretion is often not normalized on islet insulin content, which could be importantly affected in these models [48,50], and the choice of the glucose concentration range used to measure GSIS could be more extensive to reveal subtle differences.

One important parameter on the effect of maternal HF feeding on endocrine pancreas development is the period the diet is provided to the mother. Dyrskog et al. highlighted that ex vivo insulin secretion is altered only when dams receive HF diet during gestation instead of gestation and lactation. This suggests that an excess of saturated fatty acids during the gestation period negatively influences insulin secretion capacity in offspring but the same diet applied during the lactation period compensates for the negative influence during gestation [49].

The fact that the effects of maternal fat intake could persist even if excessive lipid intake is interrupted suggests that metabolic alterations are programmed and therefore could even be intergenerational. For example Graus-Nunez et al. showed that pancreatic remodeling and altered insulin sensitivity observed in the first generation of mice born from obese mothers are still present in the second generation [43]. Epigenetic modifications could explain the persistence of phenotype transmission induced by maternal nutritional intervention, but few experimental studies have been conducted so far. In vitro long-term exposure of INS-1 cells to high palmitate concentration fails to show DNA methylation of the *Ins1* promoter, unlike glucose exposure [40].

To summarize, an excess of certain types of lipids in maternal diet affect pancreas structure and function in offspring later life. Since results largely vary between studies, the timing of maternal HF feeding and the quality of fat given has to be taking into account to compare the results between studies. It also highlights the complexity of studying lipid excess consumption in human cohorts because of lipid diversity which can deeply explain interindividual variations observed in humans. In fact, certain types of lipids such as *n*-3 PUFA consumed during early pregnancy are associated with improvement in attention functions and attention deficit hyperactivity disorder, involving 2 single-nucleotide polymorphisms related with PUFA metabolism, in genes coding for glucokinase regulator and fatty acid elongase 2 [54].

#### 3.1.3. Protein Excess

Amino acids are essential for protein synthesis and other nitrogenous substances such as catecholamines, creatine, dopamine, nitric oxide, polyamines, and thyroid hormones. Several amino acids including arginine and leucine are capable of activating the mammalian target of rapamycin (mTOR) signaling responsible for protein synthesis. Arginine is a precursor for nitric oxide synthesis involved in placental angiogenesis. Arginine also directly increases placental growth to improve blood flow across the placenta, thereby increasing nutrient transfer from the mother to her fetus [55]. Despite the fact that proteins are essential for fetus development, high protein intake during pregnancy leads to an excess of circulating amino acids which are metabolized to finally produce ammonia and other metabolites such as indoles, which are toxic for embryonic and fetal survival and growth [56].

Scant evidence is available on the impact of protein excess during fetus development and growth [57,58] and clinical interventional studies are limited due to ethical restriction. In rodents, high maternal dietary protein intake, rising to 40% compared to 20% of control diet, is linked to intrauterine growth restriction (IUGR) [57]. However, the use of protein-supplemented milk to feed rat pups suffering IUGR affects their neurodevelopment and their food intake and body weight gain in adulthood [58]. The same nutritional intervention has a stimulating effect on pancreatic islet number, percentage of beta cells and plasma insulin levels, suggesting an alteration in pancreas development and glucose metabolism in adulthood [59]. In humans, high maternal protein intake is linked to a decrease in insulin-like growth factor (IGF)-II, IGF binding protein-3, and insulin concentrations in umbilical cord blood—all factors involved in placental and embryonic growth as well as nutrient transfer to the fetus [60]. Furthermore, this association between cord blood IGF-II levels and maternal protein intake was stronger in female offspring. Other studies have demonstrated important sex-specific variations between cord blood growth hormone levels and maternal protein intake, reinforcing the necessity to separately study and analyze females and males [61,62].

### 3.2. Nutrient Restriction

The most common cause of nutrient restriction is undernutrition. According to the WHO, 155 million infants are stunted globally, resulting from not enough food, a vitamin- and mineral-poor diet, inadequate childcare and disease. Maternal and fetal undernutrition accounts for over 10% of the global burden of disease. However, in Western countries, fetal undernutrition is more likely the result of a poor exchange of nutrients between the mother and the fetus through the placenta [63]. In the 1990s, Barker et al. revealed that, compared to babies born with a normal weight, infants born with a LBW as a result of IUGR are at much higher risk of developing obesity, insulin resistance, hypertension, and cardiovascular disease after they reach adulthood [6,64,65]. Epidemiological and experimental evidence accumulated over the last two decades confirmed that LBW, regardless of its etiology, is linked to a higher risk of chronic disease in adulthood. Growth-retarded newborn infants have been shown to have reduced numbers of beta cells and reduced insulin secretion [66]. In addition, this effect persists because even after a regular diet in the postnatal period, there is a reduction in the number and size of pancreatic islet cells and reduced vascularization of the endocrine pancreas [66,67,68]. This could be partially explained by epigenetic modifications that affect important genes involved in pancreas development such as Pdx-1 [69]. The most reported nutrient restriction in the literature is protein restriction for the mother, which appears to have the greatest effect on insulin secretion and glucose metabolism. 

#### 3.2.1. Carbohydrate and Lipid Restriction

Specific restriction of carbohydrates during pregnancy is a common dietary treatment for pregnant women who develop GD, which accounts for 7%–14% of all pregnancies [34,39,70]. However, the decrease in carbohydrates consumption induces a metabolic shift in favor of the use of fat utilization. The main sources of fuel are then fatty acids from dietary fat and adipose stores and ketones from dietary fat, protein, and adipose stores. Only glucose-dependent tissues continue to receive glucose from glycogenolysis and gluconeogenesis. Several teams have demonstrated a link between increased maternal lipids and patterns of fetal overgrowth [71,72]. In addition, high levels of circulating lipids and ectopic lipid storage exacerbate insulin resistance in the mother and ultimately lead an increase in blood glucose [73].

The use of low-carbohydrate diets during pregnancy has recently been extensively reviewed in the context of GD [33]. In summary, instructions for a less-restrictive carbohydrate diet, including an optimal mixture of higher quality carbohydrates with a lower glycemic index and less fat, may be preferred to a restrictive diet because of its reduced effect on metabolism.

To our knowledge, regarding the effect of maternal lipid restriction, there is no information on their potential role in the development of the fetal pancreas. In fact, maternal lipids are metabolized in the placenta or transported to the fetus for storage in fetal adipose tissues under the control of fetal insulin, particularly during the third trimester of pregnancy [74]. The storage of maternal fat should compensate for the lack of fat provided by the diet. Furthermore, a restrictive lipid diet is probably offset by an increase in carbohydrates that can be a source of newly synthesized triglycerides through glycolysis and fatty acid synthesis.

#### 3.2.2. Protein Restriction

Proteins are essential during pregnancy since they bring precursors for de novo protein synthesis and hormones, especially insulin which has an important anabolic role. Amino acids are also the main stimulus of insulin secretion during fetal life. In humans, low protein intake by the mother during gestation and lactation has several negative effects, including maternal insulin resistance, IUGR, cognitive abnormalities in offspring and pre-eclampsia [56]. Human fetuses from mothers with low protein intake have greater abdominal adiposity compared to intermediate and high protein intake, regardless of carbohydrate and lipid intake [75]. By decreasing the activity of placental 11 β-hydroxysteroid dehydrogenase, low maternal protein intake increases fetal exposure to maternal cortisol, resulting in changes in growth, blood pressure, and glucose metabolism in the fetus [76].

Several studies have demonstrated the importance of proteins during gestation and/or lactation on pancreas development using a maternal protein restriction model. A general characteristic observed in offspring is the reduction of birth weight without affecting the placenta weight [67,68,77,78,79,80]. When the low protein diet (LP; 8%) is administrated during gestation in the rat, there is a decrease in islet size and beta-cell proliferation in fetuses at 21 days of gestation, as well as a sharp reduction in the vascularization of the islet [67]. A more severe diet (5% protein) results in a similar decrease in the number, the volume and the size of the islets in male mouse offspring, with persistent effects over three generations [81]. 

Maternal protein restriction during the gestation period induces a change in islet size distribution in female rat offspring as the male is generally not studied [68]. The main observation is a significant decrease in the total islet area in low-protein group pups at PND7 with a higher number of smaller islets and the absence of large islets in adulthood. However, the number of beta cells does not seem to differ in adulthood, suggesting a compensatory response to maintain normal blood glucose level [68]. Some studies suggest that the evolution of malnutrition and/or protein restriction over time may influence the development of the pancreas differentially. Bertin et al. have specifically studied the role of energy restriction and/or protein restriction during the third part of gestation, which corresponds to the main period of pancreas development [82]. In that model, protein deficiency, but not energy restriction, induces persistent damages to pancreas development and in particular impacts beta-cell mass in female offspring. It is reassuring to note that when the offspring is fed with chow diet after weaning, glucose production, utilization, and glucose tolerance are not drastically affected in these models of malnutrition. When maternal protein restriction takes place specifically during gestation, basal insulin secretion in response to glucose is increased in isolated islets from PND36-aged male rats [83]. However, GSIS is impaired in islets from adult and aged offspring from protein-restricted mothers whatever the period of restriction. So, maternal protein restriction seems to impair GSIS in adults. Ex vivo measurement of insulin secretion on isolated pancreas shows that a low protein diet during gestation and lactation decreases insulin secretion in response to glucose, arginine or both stimuli in offspring (PND42) without muscle insulin sensitivity increase [84]. The period where the LP diet is given to the mother seems to have a real importance regarding the impact on insulin secretion. As observed for maternal HF feeding, maternal protein restriction alters pancreatic development for up to three generations, reinforcing the persistent restricted-programmed phenotype in offspring [81].

Differences observed between studies could be explained by the different amounts of protein in diets but also mainly by the timing of the nutritional intervention [82,83]. A slight disruption of insulin secretion or organization of the pancreas may not impact the mechanism of glucose tolerance under standard conditions but may result in metabolic disease when caused by high-energy diet challenge, aging or genetic origin.

## 4. Impact of Ultra-Processed Food Consumption

Ultra-processed foods are defined by food products made mostly or entirely from sugar, oils, and fats, and other substances not commonly used in culinary preparations such as preservatives, sweeteners, hydrogenated oils, modified starches, and protein isolates. These highly palatable products are also highly profitable (low-cost ingredients, long shelf-life, emphatic branding) and convenient (ready-to-consume) [85]. It is now well-established that high consumption of ultra-processed foods such as potato chips or sweets is associated with higher risk of weight gain and obesity. On the contrary, an inverse relationship exists between unprocessed/minimalized processed food consumption such as whole grains and vegetables and body weight gain [86].

The current lifestyle in Western countries fosters out-of-home and ready-to-eat meals. Several recent articles support the fact that the increase in ultra-processed food consumption has a harmful impact on human health. In a nationally representative sample of US adults participating in the National Health and Nutrition Examination Survey 2005–2014, higher consumption of ultra-processed foods has been associated with excess weight, with a more pronounced association among women [87]. In France, by a large observational prospective study on the NutriNet-Santé cohort, it has been shown that higher consumption of ultra-processed foods was associated with higher risks of cardiovascular, coronary heart, and cerebrovascular diseases [88]. Increasingly studies observe this type of association. Since obesity is a multifactorial disease, data should be adjusted on obesity-related confounding factors such as physical activity to avoid bias [86]. However, so far, very few studies take these factors into account.

Unfortunately, very limited studies are available on a direct link between mother consumption of ultra-processed foods during gestation and lactation periods and offspring outcomes. In a sample of pregnant women from a large US cohort, it has been shown that the percentage of energy intake from ultra-processed foods is associated with increased gestational weight gain and neonatal body fat [89]. Overall, studies on this topic mostly demonstrate a positive association between ultra-processed food intake and obesity but did not necessarilyy search for a link with the appearance of GD [87,90,91,92]. Maternal education and sociodemographic characteristics are positively associated with the consumption of ultra-processed foods but could be challenged with the help of health professionals offering educational interventions [91,92].

Despite new knowledge in the field of ultra-processed foods, additional studies are needed to confirm and extend current observations to the world’s population, with the majority of studies so far focusing on South American populations. However, the results of these studies recommend a limited consumption of ultra-processed foods during pregnancy and encourage the use of a healthy diet during this critical period to improve maternal and newborn health. It is now essential to validate an easy and common classification system that will be used to standardize public health studies.

## 5. Other Molecules that Can Influence Pancreas Development and Function: The Case of Xenobiotics

Xenobiotics are chemicals found in an organism and are not naturally produced or present in the body. They can be toxic to the body even at very low doses. The typical and most studied xenobiotics are pesticides and drugs, especially antibiotics.

During gestation, the placenta connects the fetus physically and biologically to the uterine wall and provides water, nutrients and dioxygen [93]. Despite its role as a barrier, some lipophilic substances pass through the placenta by passive diffusion. In addition, the ATP-binding cassette (ABC) transporters located in the placenta recognize xenobiotics and return them to the maternal circulation, thus protecting the fetus. However, these efflux proteins are sensitive to certain molecules or viruses and even epigenetic mechanisms which can modulate their activity and diminish their protective role [93,94]. The presence of pesticides residues that accompany food consumption is associated with a higher risk of impaired neurodevelopment and obesity in rodents and humans [95,96,97]. In utero exposure to environmentally relevant doses of bisphenol A modifies the development of the pancreas exclusively in young males [98]. The mass and proliferation of beta cells increase as well as the expression of the regulatory genes of the cell cycle. These changes that do not persist in adulthood may be responsible for alterations in glucose homeostasis observed later [98].

Malaria in pregnancy can deeply affect fetal development. Recent findings in a murine model of malaria-induced preterm labor show that the expression of several ABC transporters is decreased in pregnant mice, suggesting a potential modulation of placental nutrient, environmental toxin and xenobiotic biodistribution within the fetal compartment [94]. However, nothing is known on the development of the pancreas. As xenobiotics are increasingly present in our environment, it seems important to thoroughly evaluate their direct impacts on offspring and on the epigenome to prevent intergenerational effects.

## 6. Importance of Lactation

Milk provides not only the raw material necessary for growth, but also a large number of bioactive components, so far incompletely duplicated by infant formula. The beneficial effects of breastfeeding on short- and long-term health outcomes probably depend on these compounds. Although breastfeeding has been associated with a slower initial growth rate, clinical studies have shown a beneficial effect of breastfeeding on long-term neurodevelopment as observed recently in two cohorts of preterm infants, the LIFT regional cohort and the EPIPAGE nationwide French cohort [99]. The main macronutrients in milk are lipids, essential for energy metabolism, membrane construction, brain development and as precursors of signaling lipids as prostaglandins or endocannabinoids; carbohydrates for energy metabolism; and proteins for growth, modulation of immune system and defense against pathogens [100]. Lipids are the largest source of energy in breast milk, accounting for 40%–55% of the total breastmilk energy required by a healthy baby [101]. Triglycerides represent 98% of the lipid fraction and are mainly formed of newly synthesized fatty acids in the mammary gland or extracted from maternal plasma [102]. It is now well recognized that diet and maternal metabolism in humans and rodents may affect the quality and quantity of the lipid fraction of breast milk [100,102,103,104], particularly the level of *n*-3/*n*-6 PUFA [105]. Long-chain PUFA supplementation in rat dams increases the *n*-3/*n*-6 PUFA levels in maternal milk and pup brains [106]. Human milk proteins are essentially constituted by fractions of whey and casein fractions produced by lactocytes at 80%–90% [102]. Hormones such as leptin and insulin are also present in breast milk.

The protein concentration of milk depends on the amount of milk produced and the body mass index of the mother, but not on the diet [100]. Breast milk contains more lactose than rodent milk, which is richer in lipids. In addition to diet, milk composition varies during breastfeeding. Average levels of protein and lactose decrease over time, while lipid levels increase progressively with breastfeeding [102] and the same is observed in rodents.

Breast milk can also contain xenobiotics although their presence is low in exclusively breastfed infants [107] with the exception of contaminants originated from a particular exposition to the mother, as lead for example, or during drug treatment. However, it is important not to deprive babies of the health benefits of breastfeeding. Biological monitoring of breast milk contaminants could be set up to monitor infant exposure and assess the benefit/risk ratio in specific pollutions. 

## 7. Conclusions

Nowadays, the consumption of a diet rich in energy is common in Western countries. The high availability of food, the increased consumption of ultra-processed foods poor in nutrients and decreased physical activity are important factors in deregulating the energy balance. During the gestation period, an unbalanced diet due to excessive or restrictive consumption of nutrients directly affects fetal growth but also the metabolism of the mother, which can lead to metabolic alteration in the future adult. Since metabolic pathways are often interdependent, the absence of certain precursors in the diet can alter the availability of other molecules needed by the fetus. *It is therefore extremely complicated to study the role of one type of nutrient without considering its impact on other metabolic pathways and it is therefore more advantageous to study diet effects rather than separate nutrients.*

The breastfeeding period should not be overlooked as breast milk is strongly influenced by maternal nutrition and environment. The presence of xenobiotics in milk should be researched to study postnatal exposure to these molecules which may affect offspring epigenome. In addition, despite the improved quality of infant formula, they are still of less benefit than breast milk. A balanced diet during the gestation and lactation periods must be encouraged by specific diet recommendations and may prevent adverse effects on metabolism, which could be on the contrary aggravated by an unbalanced diet in adulthood. Recommendations on best food quality intake during gestation and lactation periods could prevent an imbalanced supply of nutrients to fetuses and babies and limit the pre- and postnatal exposure to disruptive molecules.

Recent publications attempt to separate the observations made in female offspring and male offspring. However, the sex of the children is not clearly documented in many publications. Due to sex hormones, the metabolism of male and female children may react differently to the mother’s nutritional intake and a clear sexual dimorphism appears for example in terms of glucose metabolism. Sexual polymorphism deserves better consideration when exploring diet impact on pathophysiological processes.

The transgenerational effects observed in experimental studies support the involvement of epigenetic modifications on genes important for pancreas development and glucose metabolism which explain transmission to future generations. However, it is necessary to consolidate experimental observations using a human cohort follow up despite the difficulty in exploring pancreatic function in babies or infants.

## Figures and Tables

**Figure 1 nutrients-11-02708-f001:**
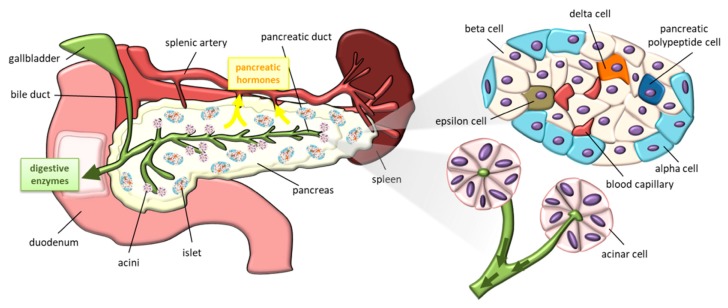
Schematic representation of pancreas (left side), pancreatic islet and acini (right side).

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
