# Peer review of "Effects of Nutrient Intake during Pregnancy and Lactation on the Endocrine Pancreas of the Offspring"

_nutrients, 2019, doi:10.3390/nu11112708_

Round 1
Reviewer 1 Report
The aim of the paper is to highlight the influence of nutrient intake during gestation and lactation on pancreas development. Its main contributions: an unbalanced diet due to excessive or restricted consumption of nutrients directly affects fetal growth. Sex may react differently to the mother´s nutritional intake. Transgenerational effects observed in experimental amino acids.
Areas of strength: Nicely written and clearly supported that nutrition during pregnancy is related with child outcomes.
Areas of weakness: Very general and long introduction, NO DATA discussed ON Ultra-processed Food Consumption and pregnancy outcomes neither on xenobiotics. No genetic associations in the mother with food intake discussed. No clear conclusions and practical recommendations, many conflicting mechanisms or outcomes.
Author Response
First, we thank the reviewer #1 for their helpful comments which improve our manuscript.
Reviewer #1
Comments and Suggestions for Authors
The aim of the paper is to highlight the influence of nutrient intake during gestation and lactation on pancreas development. Its main contributions: an unbalanced diet due to excessive or restricted consumption of nutrients directly affects fetal growth. Sex may react differently to the mother´s nutritional intake. Transgenerational effects observed in experimental amino acids.
Areas of strength: Nicely written and clearly supported that nutrition during pregnancy is related with child outcomes.
Areas of weakness: Very general and long introduction, NO DATA discussed ON Ultra-processed Food Consumption and pregnancy outcomes neither on xenobiotics. No genetic associations in the mother with food intake discussed. No clear conclusions and practical recommendations, many conflicting mechanisms or outcomes.
Response for reviewer #1
We maintained the 2 parts, sections 1 and 2 since they bring basic knowledge and references to understand pancreas development and functions.
We add a chapter on the role of ultra-processed food consumption during gestation and lactation. Although no experimental data are available on that topic, few clinical studies present some interesting data that are now presented. The same has been done for xenobiotics. Some of their effects are now presented through the use of maternal milk.
We choose not to discuss extensively genetic associations and mother food intake as our aim was to focus on pancreas development and function and we did not found a lot of literature on that specific topic.
The conclusion has been reorganized in order to propose practical recommendations and future research directions.
Reviewer 2 Report
This is a review about effects of maternal nutrient intake on the pancreatic function of the offspring.
There is a slight bias toward basic data. Especially 3.1.2. Lipid excess section is controversial.
Some human epidemiological data or information of epigenetic analysis of animal model should be described in 3.1.2. section. As mentioned in Conclusion section, epigenetics is important for understanding these areas of study. Subsequent research using human subjects such as peripheral blood or saliva may lead to next clinical findings of human metabolic disease.
Some references are needed.
Line 206, GD or obesity increases risk of developing T2D in offspring.
Line 224, HF feeding reduces whole-body insulin sensitivity in offspring.
Line 302, Few evidences are available on the impact of protein excess during fetus development and growth.
Line 334, high levels of circulating lipids exacerbate insulin resistance in the mother.
Author Response
First, we thank both reviewers for their helpful comments which improve our manuscript.
Reviewer #2
Comments and Suggestions for Authors
This is a review about effects of maternal nutrient intake on the pancreatic function of the offspring.
There is a slight bias toward basic data. Especially 3.1.2. Lipid excess section is controversial.
Some human epidemiological data or information of epigenetic analysis of animal model should be described in 3.1.2. section. As mentioned in Conclusion section, epigenetics is important for understanding these areas of study. Subsequent research using human subjects such as peripheral blood or saliva may lead to next clinical findings of human metabolic disease.
Some references are needed.
Line 206, GD or obesity increases risk of developing T2D in offspring.
Line 224, HF feeding reduces whole-body insulin sensitivity in offspring.
Line 302, Few evidences are available on the impact of protein excess during fetus development and growth.
Line 334, high levels of circulating lipids exacerbate insulin resistance in the mother.
Response for reviewer #2
We tried to simplify the 3.1.2. section and add information about humans and epigenetics.
New references have been added lines 209, 231, 326, 367 and highlighted in yellow.
Round 2
Reviewer 2 Report
Thank you for your rivision.